# Effects of Pendrin Protein in Nasal Epithelial Cells on Mucin Production in the Context of Type 2 Inflammation

**DOI:** 10.3390/jpm13030502

**Published:** 2023-03-10

**Authors:** Nongping Zhong, Honghui Ai, Wei Zhong, Xiaoyan Huang, Kai Wang, Qing Luo, Jieqing Yu

**Affiliations:** 1Department of Otorhinolaryngology Head and Neck Surgery, The First Affiliated Hospital of Nanchang University, Nanchang 330006, China; 2Department of Otorhinolaryngology, The 908th Hospital of Chinese People’s Liberation Army Joint Logistic Support Force, Nanchang 330006, China; 3Department of Allergy, The First Affiliated Hospital of Nanchang University, Nanchang 330006, China; 4Jiangxi Otorhinolaryngology Head and Neck Surgery Institute, The First Affiliated Hospital of Nanchang University, Nanchang 330006, China

**Keywords:** chronic rhinosinusitis, mucin, pendrin, type 2 inflammation

## Abstract

Background: Chronic rhinosinusitis (CRS) is a heterogeneous disease. The pathogenesis of chronic sinusitis is still unclear; however, the nasal cavity and paranasal sinuses are commonly affected by type 2 inflammation, which is caused by Th2 cytokines such as interleukin (IL)-5, IL-4, and IL-13. Previous studies have shown that pendrin promotes local infiltration of neutrophils through the production of human neutrophil elastase (HNE), which is essential for the secretion of mucin 5AC (MUC5AC) in chronic inflammatory diseases of the lower respiratory tract. This study investigated pendrin expression and its relationship to mucin in type 2 inflammation. Methods: A total of 40 patients (10 CRS patients with nasal polyps,10 CRS patients without nasal polyps, and 20 nasal septum deviation patients) were included in this study and were divided into the CRS group and the NC group. A normal nasal mucosa tissue culture model was established in vitro. IL-13 was used to stimulate primary cultures of human nasal epithelial cells (HNECs). Western blot (WB), enzyme-linked immunosorbent assay (ELISA), and quantitative real-time polymerase chain reaction (qRT-PCR) were used to detect the expression of pendrin, MUC5AC, and MUC5B. After transfecting HNECs with siRNA pendrin or negative control (NC), EGF receptor (EGFR), HNE, MUC5AC, and MUC5B expression were analyzed using qRT-PCR, WB, or ELISA in terms of their relationships with pendrin. Pendrin expression in the tissue was also analyzed. Results: After IL-13 stimulation, pendrin, MUC5AC, and MUC5B expression levels were upregulated; the optimal concentration of IL-13 was 50 ng/mL. The expression levels of HNE, EGFR, MUC5AC, and MUC5B were downregulated after transfection with siRNA pendrin-1650. Pendrin expression in the NC group was lower than in the CRS group. Conclusion: IL-13 is implicated in the inflammation of nasal mucosa, and pendrin is closely related to the excessive secretion of mucin. The expression of mucin is downregulated after transfection with siRNA pendrin. There is a positive relationship between pendrin and EFGR/HNE. Moreover, pendrin plays an important role in type 2 inflammation.

## 1. Introduction

Chronic rhinosinusitis (CRS) is a common disease classified as primary chronic rhinosinusitis and secondary chronic rhinosinusitis, which affects 5–12% of the general population [1]. CRS has a significant impact on the quality of life of patients and imposes a large social burden [1,2]. The prognosis of CRS is affected by several factors, especially the pathophysiological mechanism. The pathogenesis of CRS involves the epithelial cell barrier, tissue remodeling, immunology, and inflammation. According to the European Position Paper on Rhinosinusitis and Nasal Polyps (EPOS) 2020 guidelines, primary CRS can be classified as diffuse or limited, both distinguishable as type 2 and non-type 2 inflammatory conditions. ECRS (Eosinophilic CRS) and CRSwNP (CRS with nasal polyps) can be classified under type 2 inflammation [3]. Inflammation is considered to play a key role in the persistence of CRS [4]. The long-term prognosis of CRS remains poor for the recurrence is high [5,6,7], especially in CRS due to type 2 inflammation, and the underlying mechanism must be identified to improve clinical outcomes. Mucus is mainly secreted by goblet cells and submucosal mucous glands. Playing an important role in airway defense and repair, the mucus barrier is a key factor in the innate defense of the airway; increased mucus concentration and mucin concentration affect mucus cilia clearance, leading to mucus buildup in the airway and increasing the risk of chronic inflammation. Hypersecretion of mucus is one of the main characteristics of CRS [8], usually caused by the upregulation of mucin. Mucins (MUC) are the major components of mucus. Nine major mucins are detected in the airway: mucin 1 (MUC1), MUC2, MUC4, MUC5AC, MUC5B, MUC7, MUC8, MUC11, and MUCl3 [9]. MUC5AC (expressed in goblet cells in the airway epithelium) and MUC5B (mainly expressed in submucosal glands) are the predominant MUC types in normal airways and some respiratory diseases [10,11]. Furthermore, a previous study showed that MUC5AC expression was upregulated in the sinus mucosa of CRS patients compared with those with normal sinus mucosa [12]. Using qPCR and immunohistochemical analysis, American scholars Kim et al. demonstrated significantly higher expression of MUC5AC and MUC5B in CRS patients compared with those with normal tissue, expressed by different mucus-secreting cell types in sinus mucosa [13]. These studies thus demonstrate that MUCs may play an important role in the pathogenesis of CRS. Previously, we showed that human neutrophil elastase (HNE) induces MUC5AC overexpression in primary human nasal epithelial cells (HNECs) [14], further illustrating the pathophysiological mechanism of increased secretion of MUC5AC in CRS.

Pendrin protein is a soluble transporter. An important part of the family 26A (SLC26A) member, encoded by the SLC26A4 gene, can mediate cell chloride ion, iodine dissociation, and the exchange of various anions such as zinc and bicarbonate ions to maintain a steady state of medium ions in the tissue. Pendrin is primarily localized in the inner ear, thyroid, and kidney [15,16,17]; however, under pathological conditions, airway epithelium cells can also secrete pendrin [18], a key mediator of mucus production by airway epithelial cells. Pendrin overexpression led to systemic pathophysiological changes in the airways of a mouse model of airway disease [19]. Overexpression of pendrin induces airway hyper-responsiveness, mucus exudation, and neutrophil infiltration in mouse lung tissue. Neutrophil elastase released from lung-infiltrating neutrophils enhances mucin gene expression through transcriptional and post-transcriptional mechanisms: neutrophil elastase activates EGFR signaling via several inflammatory mediators (including TGF-) and then activates the MAP kinase pathway and the above transcription factors, thereby upregulating the MUC5AC gene. Additionally, MUC5AC is upregulated in airway lavage fluid [20]. Studies have shown that pendrin is a novel IL-13/IL-4-induced molecule. Therefore, we speculate that pendrin in the airway epithelia can serve as a specific target in the treatment of type 2 inflammation [21].

Combining the results of the above studies, we explored that with the stimulation of some inflammatory factors and other pathogenic factors, pendrin may enhance the infiltration of NEU in the nasal cavity and paranasal sinus mucosa, and then promote the secretion and expression of mucin in epithelial and glandular tissues through EGFR signal, which led to the occurrence and development of CRS.

In this study, through interleukin (IL)-13 stimulation, and pendrin siRNA in HNECs, we analyzed the expression of HNE, EGF receptor (EGFR), MUC5AC, and MUC5B, and investigated the pendrin–EGFR–MUC pathway in CRS.

## 2. Materials and Methods

### 2.1. Subjects

Samples of nasal mucosa were collected from patients undergoing functional endoscopic surgery. A total of 40 patients (10 CRS patients with nasal polyps, 10 CRS patients without nasal polyps, and 20 nasal septum deviation patients) were included in this study from The First Affiliated Hospital of Nanchang University. The 40 patients were divided into the CRS group and the NC (negative control) group. The CRS diagnoses were based on the patients’ clinical histories and anterior rhinoscopy/nasal endoscopy/computed tomography results, in accordance with EPOS 2020 guidelines [1]. The exclusion criteria were a diagnosis of fungal sinusitis, asthma, aspirin intolerance, and cystic fibrosis. All patients stopped oral corticosteroid consumption for at least 1 month, and topical application for at least 2 weeks, before surgery. The patients did not take any other relevant medications.

The study was approved by the Medical Research Ethics Committee of The First Affiliated Hospital of Nanchang University, and each participant has signed the informed consent form.

### 2.2. Primary Culture and Stimulation of Human Nasal Epithelial Cells and Treatment and Transfection with siRNA

HNECs were cultured using a method described in a previous study [12]. Briefly, specimens were flushed with sterile saline and washed two to three times with phosphate-buffered saline (PBS). Digestive enzyme mixture (MEM 10 mL, pancreatic enzymes 1.4 mg/mL, and Dnase 0.01 mg/mL) was added to the specimen tubes. Fetal bovine serum (FBS) was then added to stop the digestion, followed by the addition of minimum essential medium (MEM). The cell suspension was centrifuged. The supernatant was aspirated. Dulbecco’s modified Eagle’s medium (DMEM)/F12 complete medium was made into a pellet. The resulting cell suspension was incubated (37 °C, 5% CO_2_), centrifuged, and cultured in bronchial epithelial cell growth medium (BEGM) (37 °C, 5% CO_2_).

After the cells reached 80–90% confluence, they were stimulated with different concentrations of recombinant IL-13 (0, 20, 50, 100, 250, and 500 ng/mL) over a 24 h period. The cell lysates for protein are as follows. Remove the culture medium and drugs, gently rinse the adherent cells with precooled PBS for 3 times, pay attention to gentleness to prevent the cells from falling off, suck the remaining PBS solution in the culture dish for the last time, and discard it, add protein lysate and phosphorylated protease inhibitor, blow repeatedly, put them on the ice for 10 min, and then use the cell scraper to scrape the cells and lysate, and transfer them to 1.5 mL EP tube, blow repeatedly to avoid gas. Bubble is generated, but attention should be paid to avoid the generation of foam, and it should be placed on ice to continue cracking for 20 min. Put the EP tube into a precooled 4 °C centrifuge, 12,000× *g*, centrifuge for 10 min, and take the supernatant.

The cell lysates for RNA are as follows. The cells were washed by D-hankS solution three times. After trypsin digestion, the cell suspension was collected with a 15 mL centrifuge tube. Total RNA was extracted from cells with Trizol™ reagent (Cat No: 15596026, Invitrogen, Thermo Fisher Scientific, Waltham, MA, USA).

The cell lysates were collected and MUC5AC, MUC5B, and pendrin mRNA expression were determined by qPCR. MUC5AC and MUC5B expression in the cell supernatant was detected by enzyme-linked immunosorbent assay (ELISA). The protein expression of pendrin was evaluated using Western blot (WB) analysis.

Pendrin (si pendrin) and a non-targeting siRNA (siCTRL) were synthesized by Sangon Biotech (Shanghai, China) and transfected into target cells using Lipofectamine 2000 (Invitrogen, Waltham, MA, USA). SiNRA are as follows: **FAM NC** (Sense UUCU CCGAACGUGUCACG UTT, Antisense ACGUGACACGUUCGGAGAATT), **NC** (Sense UUCUCCGA CGU GUC ACG UTT, Antisense ACGUGACACGUUC GGAGAAT T), **Pendrin-homo-373** (Sense GCUGCAGUUGCUCAAGAAATT, Antisense UUUCUUG AG CAACUGCAGCTT), **Pendrin-homo-920** (Sense GCUAAAGAUCGU GCUCAAU TT, Antisense AUUGAGCACGAUCUUUAGCTT) and **Pendrin-homo-1650** (Sense GCUGUUAUCUGGGUGUUUATT, Antisense UAAACACCCAGAUAACAGCTT).

### 2.3. Quantitative Real-Time Transcription-Polymerase Chain Reaction (qRT-PCR)

A qRT-PCR assay was performed as described by Ye [12]. Total RNA was extracted from nasal tissue using TRIzol reagent (15596-026; Invitrogen). Reverse transcription (RT) was performed, and cDNA was synthesized and reverse-transcribed using a Revert Aid First Strand cDNA Synthesis Kit (#K1622; Thermo Fisher Scientific, Waltham, MA, USA) and a Fast Start Universal SYBR Green Master Kit (04913914001; Roche Holding AG, Basel, Switzerland) for quantitative PCR. The primers for PCR were as follows: β-actin, forward: 5′-GTGACGTTGACATCCGTAAAGA-3′, reverse:

5′-GTAACAGTCCGCCTAGAAGCAC-3′; MUC5AC, forward: 5′-ACCTGTGACAGCAGGATGTG-3′, reverse: 5′-ACTCGCAGTCTCCGTTGAAG-3′; MUC5B, forward: 5′-TCCCACTATTCCACCTTTGACG-3′, reverse: 5′-CCAGGTAGAGGCTGAGATTCCC-3′; HNE, forward: 5′-GCGTGGCGAATGTAAACGTC-3′, reverse: 5′-ACCCGTTGAGCTGGAGAATC-3′; EGFR, forward: 5′-GCCAAGGCACGAGTAACAAGC-3′, reverse: 5′-GGGCAATGAGGACATAACCAGC-3′.

### 2.4. Enzyme-Linked Immunosorbent Assay

Cell supernatants were collected and subjected to ELISA to determine the levels of human MUC5AC and MUC5B secretion using a specific ELISA assay kit (SEA756Hu 96T; Cloud-Clone Corp., Wuhan, China) and (SEA684Mu 96T; Cloud-Clone Corp., Wuhan, China) according to the manufacturer’s instructions.

### 2.5. Western Blot Analysis

WB was performed in accordance with the method described by Luo et al. [10]. Total cell extracts were resolved by sodium dodecyl sulfate 10% polyacrylamide gel electrophoresis. The protein was then electrophoretically transferred to polyvinylidene fluoride membranes. The membranes were incubated overnight at 4 °C with diluted (1:1000) rabbit anti-human pendrin (bs-19817R; Bioss, Beijing, China), rabbit antihuman HNE (ab68672; Abcam, Cambridge, UK), rabbit anti-human EGFR (bs-0405R; Bioss), β-actin polyclonal antibodies (GB13001-1; Servicebio, Wuhan, China), and glyceraldehyde-3-phosphate dehydrogenase polyclonal antibodies (GB13002-m-1; Servicebio, Wuhan, China). Membranes were then washed and incubated in goat anti-rabbit antibodies (GB23303; Servicebio, Wuhan, China) for 1 h. Finally, membranes were washed three times with Tris-buffered saline–Tween. Under the condition of avoiding light, the A and B solution (Beyotime Biotech, Shanghai, China) in the luminescent liquid reagent shall be mixed evenly by 1:1. After incubation for an appropriate time, membranes were exposed in the chemiluminescence instrument (Bio-Rad company, Hercules, CA, USA).

### 2.6. Statistical Analysis

Statistical analysis was performed using SPSS 22.0 (IBM Corp., Armonk, NY, USA). Expression data are presented as dot plots of the median and interquartile range. For the in vitro experiments, cell culture data are presented as means ± standard error of the mean (SEM). Data were analyzed using GraphPad Prism 8 (GraphPad Software Inc., San Diego, CA, USA). The differences between the CRS and negative control (NC) groups were examined for significance using Student’s *t*-test. In all analyses, *p* < 0.05 was considered statistically significant.

## 3. Results

### 3.1. IL-13 Upregulates Pendrin, MUC5AC, and MUC5B Expression in HNECs

At the tissue level, pendrin expression was higher in the CRSsNP and CRSwNP than NC group (*p* = 0.0090 and *p* = 0.0019, respectively); the difference in pendrin expression between the CRSsNP and CRSwNP groups was not statistically significant (*p* = 0.2925, Figure 1). As in our previous study [22], MUC was upregulated in the CRS compared with NC group. (Lanes 1–3 are the samples from the NC group, 4–6 are the samples from the CRSsNP patients and 7–9 are the samples from the CRSwNP patients.

To investigate the role of IL-13 in mucous secretion, we treated primary HNECs with different concentrations of IL-13 and cultured them for 24 h. With IL-13 stimulation at concentrations < 50 ng/mL, pendrin, MUC5AC, and MUC5B mRNA were significantly upregulated in primary cultured HNECs (Figure 2). The expression levels of pendrin, MUC5AC, and MUC5B peaked when the IL-13 concentration reached 50 ng/mL (Figure 3). RT-PCR and WB analysis revealed that, when HNECs were treated with 50 ng/mL, MUC and pendrin showed the most obvious increase. These findings indicate that with IL-13 stimulation, pendrin and MUC play key roles in the pathogenesis of sinusitis. These results suggest a functional correlation between pendrin and MUC.

### 3.2. Pendrin Influences MUC5AC and MUC5B Expression

HNECs were transfected with NC, pendrin-373, pendrin-920, pendrin-1650, and FAM NC, with IL-13 (50 ng/mL) stimulation. FAM NC was used to evaluate transfection efficiency. The results showed that after the transfection of siRNA pendrin-1650, both the mRNA and protein levels of pendrin decreased (Figure 4 and Figure 5). SiRNA is a gene interference entity capable of downregulating the expression of pendrin. After transfection of siRNA pendrin-1650 and IL-13 stimulation, ELISA and qRT-PCR analyses indicated that MUC5A and MUC5B were downregulated (Figure 6). Thus, pendrin inhibition could potentially ameliorate the hypersecretion of MUC5AC and MUC55B in primary HNECs.

### 3.3. Knockdown of Pendrin in HNECs Downregulates HNE and EGFR Expression

The methods of culturing and transfecting HNECs were described earlier. According to qRT-PCR and WB analyses, mRNA and protein levels of HNE and EGFR expression decreased, with pendrin-1650 being the most efficient siRNA (Figure 7). These results indicate that pendrin may affect the mucin expression of HNECs by activating HNE/EGFR, in line with the results of previous studies [23,24].

## 4. Discussion

CRS is a common and complex disease that affects the quality of life. The main histopathological manifestations of chronic sinusitis are mucosal epithelial thickening, goblet cell proliferation and/or metaplasia, gland hyperplasia, and edema [21]. MUC5AC is mainly expressed in goblet cells secreting mucus in the surface epithelium of the respiratory tract and is an important component of airway mucus [25].

However, in our previous research, we found that HNE induces the upregulation of MUC5AC in primary HNECs [14]. Zhao et al. also showed that mucin expression in vivo is mediated by a cascade involving the HNE–TACE–EGFR signaling pathway [26]. Moreover, HNE and EGFR may play an important role in MUC secretion [27].

It has been reported that several inflammatory factors, including IL-4, IL-8, IL-6, IL-13, and IFN-γ, are significantly increased in the mucosa of CRS patients, and that secretion is progressive [28]. Many studies have reported increased levels of IL-4, IL-5, IL-13, IL-1b, and tumor necrosis factor alpha in sinonasal mucosa from CRS patients with and without NPs [29,30]. However, IL-13 is more closely correlated with type 2 inflammation, as described in EPOS 2020. Considering this, we used IL-13 to stimulate HNECs and explore the mechanism of type 2 inflammation in CRS.

Pendrin is mainly expressed in the inner ear, thyroid, and kidney; however, recent studies showed that it is also expressed in airway diseases [31]. One study showed that the mRNA expression of pendrin in human primary tracheal epithelial cells stimulated by IL-4 increased by 23 times [32]. Under the action of some microorganisms and/or other factors (such as rhinovirus, IFN-γ, or an allergen), pendrin can exacerbate asthma, and its expression level is increased by about 4.9 times; this corresponds to alterations of airway surface liquid [18]. Kuperman found increased pendrin expression in three mouse asthma models, along with marked IL-13 upregulation [19]. In another study, with ovalbumin stimulation, pendrin knockout mice showed lower airway hyper-responsiveness and eosinophil levels, and less inflammation, compared with wild-type mice [18]. In our previous study, we found higher pendrin expression in normal than CRS tissue (Figure 7), and a positive correlation between the expression of pendrin and MUC5AC [22]. To explore the function of pendrin in type 2 inflammation, we stimulated HNECs with IL-13, in which MUC5AC, MUC5B, and pendrin expression were upregulated. The results showed that IL-13 can influence pendrin, MUC5AC, and MUC5B expression at the levels of transcription and translation.

The expression of EGFR is low in the normal respiratory epithelium; however, when activated by HNE, EGFR expression is enhanced, which promotes cell metaplasia and mucus secretion, especially the expression of MUC [33]. The secretion of MUC5AC can be stimulated by HNE [33,34,35]. Considering previous studies [36,37,38], EGFR may play an important role in MUC synthesis in CRS. EGFR is activated by HNE as phosphorylated EGFR (pEGFR) and induces MUC5AC expression. For this reason, we used IL-13-stimulating HNCECs and transfection with siRNA pendrin to reveal that the expression of EGFR, HNE, MUC5AC, and MUC5B was downregulated. Our study shows that upregulated pendrin expression leads to MUC secretion in type 2 inflammation, which is induced by IL-13; thus, MUC plays an important role in the pathogenesis of CRS.

With IL-13 stimulation, pendrin may promote the secretion and expression of mucin in the epithelium and glands through EGFR signaling, which can lead to the onset of type 2 inflammation.

There are still some limitations in our study. First, we did not use the EGFR pathway blocker to explore the relations between EGFR and MUC. Second, we lacked animal models. Experiments in vivo should be carried out to further explore the function of these genes in CRS.

## 5. Conclusions

In our study, stimulation of HNECs with IL-13 resulted in upregulated MUC5AC, MUC5B, and pendrin expression at the levels of transcription and translation. In addition, the knockdown of pendrin inhibited MUC secretion. Thus, the results of our study indicate that targeting pendrin has the potential as a therapeutic strategy for mucin hypersecretion in CRS patients.

## Figures and Tables

**Figure 1 jpm-13-00502-f001:**
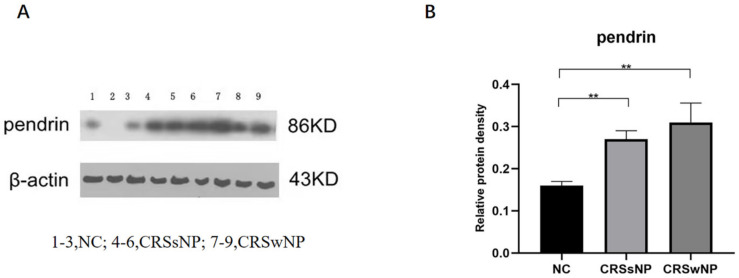
Pendrin protein expression in nasal samples revealed by Western blot (WB). (**A**) is the pendrin expression in tissuse, (**B**) is the density of (**A**) pendrin. ** represents *p* < 0.01.

**Figure 2 jpm-13-00502-f002:**
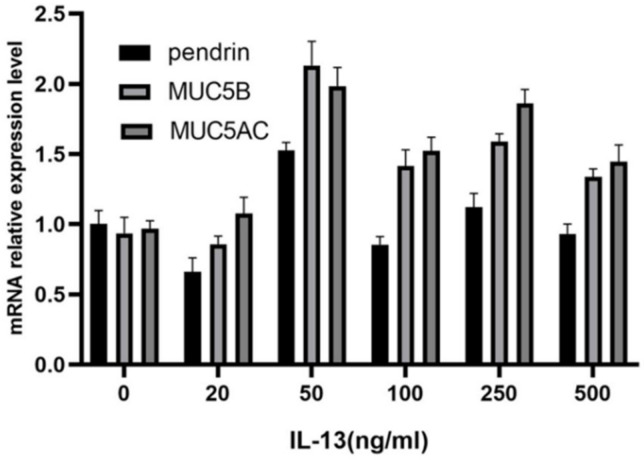
Interleukin-13 (IL-13) induces mucin 5AC (MUC5AC), MUC5B, and pendrin mRNA expression in primary human nasal epithelial cells (HNECs). The IL-13 concentrations applied were 0, 20, 50, 100, 250, and 500 ng/mL.

**Figure 3 jpm-13-00502-f003:**
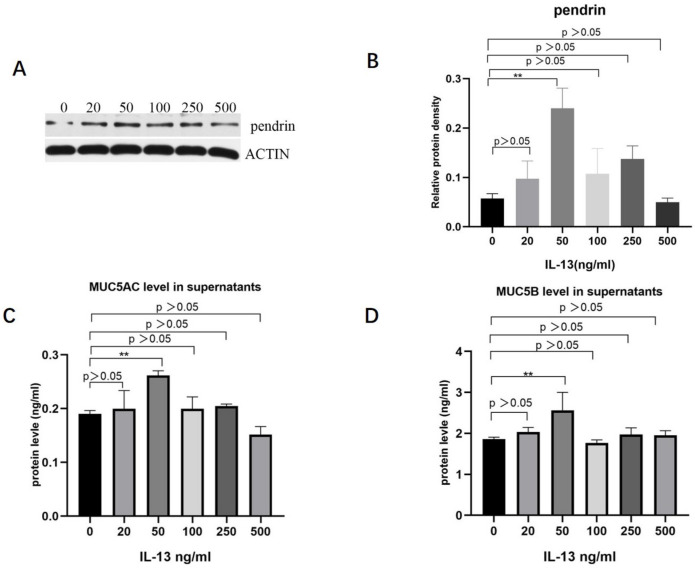
With stimulation by IL-13 (0, 20, 50, 100, 250, and 500 ng/mL), MUC5AC, MUC5B, and pendrin expression are upregulated in primary HNECs. (**A**,**B**) Show pendrin revealed by WB. (**C**,**D**) Enzyme-linked immunosorbent assay (ELISA) results showing IL-13-induced MUC5AC and MUC5B expression. ** represents *p* < 0.01.

**Figure 4 jpm-13-00502-f004:**
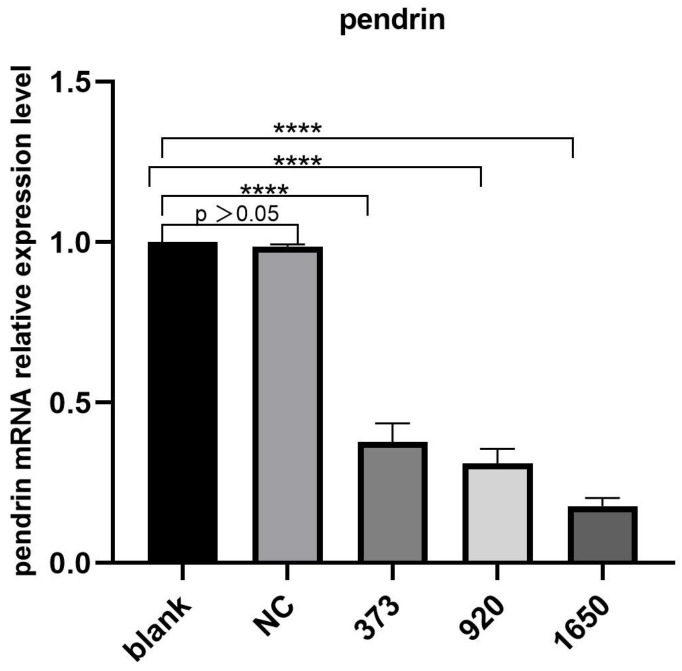
After transfection with siRNA pendrin (NC, pendrin-373, pendrin-920, pendrin-1650, and FAM NC), the mRNA expression of pendrin was downregulated. **** represents *p* < 0.0001.

**Figure 5 jpm-13-00502-f005:**
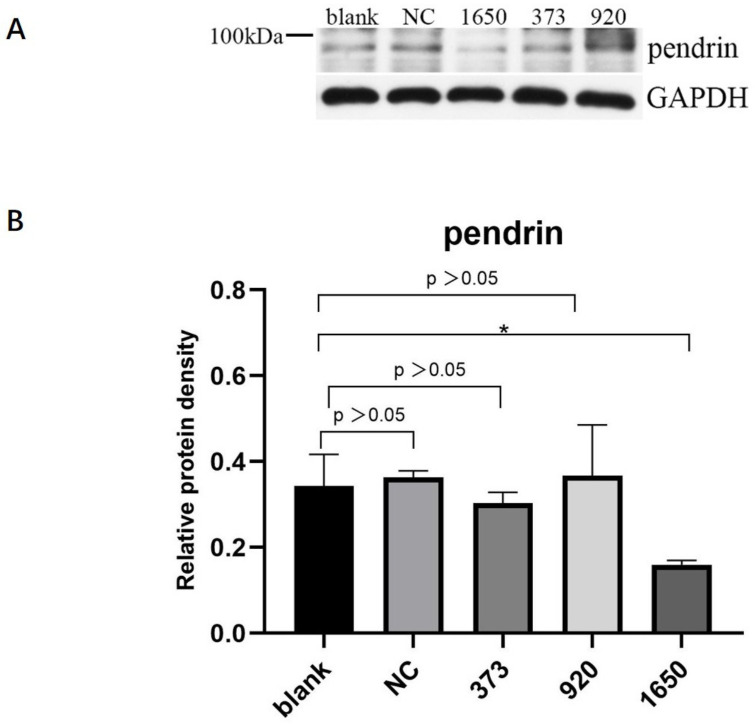
After transfection of HNECs with siRNA pendrin (NC, pendrin-373, pendrin-920, pendrin-1650, and FAM NC), the protein expression of pendrin was downregulated, especially siRNA pendrin-1650. (**A**) is the protein of pendrin after transfection siRNA pendrin, (**B**) is the density of (**A**) pendrin. * represents *p* < 0.05.

**Figure 6 jpm-13-00502-f006:**
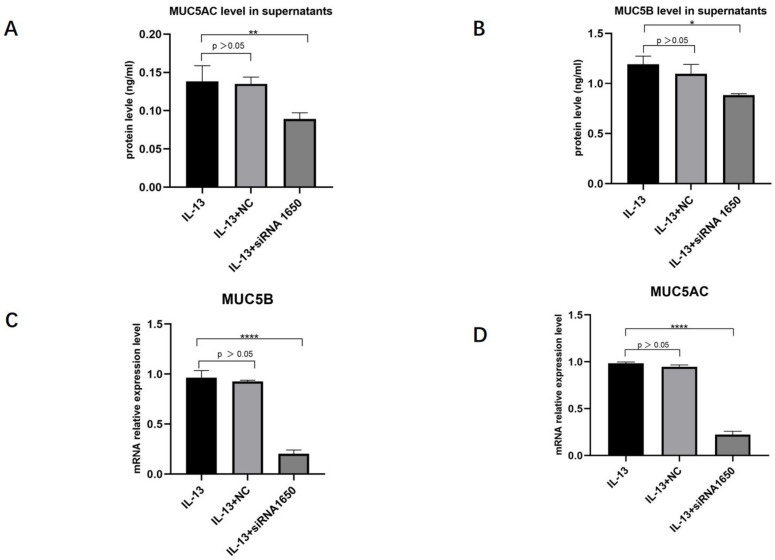
AfterIL-13 (50 ng/mL) stimulation and transfection with siRNA pendrin-1650, FAM NC, Western blot, and qRT-PCR revealed that the expression levels of MU5AC (**A**,**C**) and MUC5B (**B**,**D**) were downregulated. * represents *p* < 0.05, ** represents *p* < 0.01, **** represents *p* < 0.0001.

**Figure 7 jpm-13-00502-f007:**
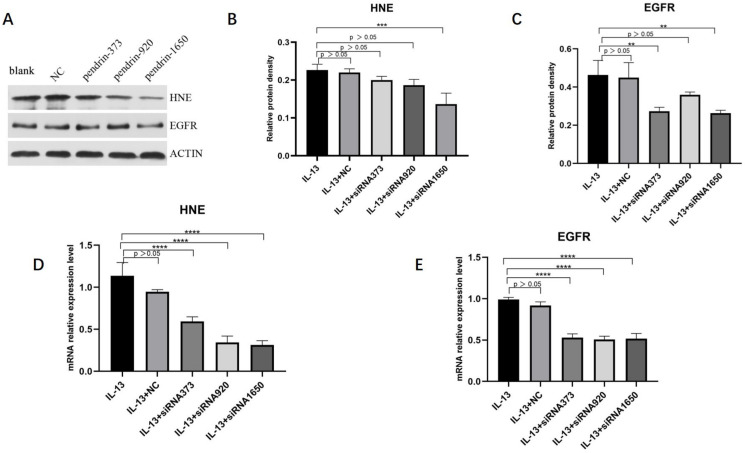
After IL-13 (50 ng/mL) stimulation and transfection with siRNA pendrin (NC, pendrin-373, pendrin-920, pendrin-1650, and FAM NC), WB and qRT-PCR showed that the expression levels of HNE and EGFR were downregulated. (**A**) is the protein expression of HNE, EGFR and ACTIN, (**B**,**C**) is the density of (**A**) HNE and EGFR. (**D**,**E**) is the mRNA expression of HNE and EGFR after transfection with siRNA pendrin. ** represents *p* < 0.01, *** represents *p* < 0.001, **** represents *p* < 0.0001.

## Data Availability

All data generated or analyzed during this study are included in this article. Further inquiries can be directed to the corresponding author.

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
