# Peer review of "Effects of Pendrin Protein in Nasal Epithelial Cells on Mucin Production in the Context of Type 2 Inflammation"

_jpm, 2023, doi:10.3390/jpm13030502_

Round 1
Reviewer 1 Report
This is an interesting paper that aims to describe pathophysiological features of CRS. However, some detailes need to be clarified:
Line 16: 'the nasal cavity is commonly afected by' - the article is dedicated to CRS (when paranasal sinuses are involved), however this line refers to nasal cavity. Please rephrase.
Lines 38-41: Authors provide a CRS classification that is a little bit outdated. Please use the classificaiton as published in EPOS 2020 (i.e. primary vs secondary etc).
Lines 46-47: ''The long-term prognosis of CRS remains poor' - please provide some references. With current treatment methods, inlcuding biologicals, is this really the case?
Line 81-82: Please provide the name of the institution that has issued the informed consents.
Line 74 states that 20 subjects with CRS were included. Please elaborate how many of those were with and how many without polyps.
Line 21 states that ''A normal nasal mucosa 21 tissue culture model was established in vitro''. However further in the discussion authors mention that patients with CRS wer eincluded. Please clarify.
Author Response
Dear Editors,
Thank you for your letter and for the reviewers’ comments concerning our manuscript entitled “Effects of pendrin protein in nasal epithelial cells on mucin production in the context of type 2 inflammation” (ID: jpm-2164344). Those comments are all valuable and very helpful for revising and improving our paper and, as well as the important guiding significance to our researches. We have studied comments carefully and have made correction which we hope meet with approval. Revised portion are marked in red in the manuscript. The main corrections in the paper and the responds to the reviewer’s comments are as flowing:
Reviewer: 1
This is an interesting paper that aims to describe pathophysiological features of CRS.
However, some detailes need to be clarified:
(1)Line 16: 'the nasal cavity is commonly afected by' - the article is dedicated to CRS (when paranasal sinuses are involved), however this line refers to nasal cavity. Please rephrase.
Response:We agreed with you. We have made correction according to the Reviewer’s comments. And we have revised this part in the manuscript in Line 19.
- Lines 38-41: Authors provide a CRS classification that is a little bit outdated. Please use the classificaiton as published in EPOS 2020 (i.e. primary vs secondary etc).
Response: According to EPOS 2020, CRS is classfied as primary chronic rhinosinusitis and secondary chronic rhinosinusitis.
(3)Lines 46-47: ''The long-term prognosis of CRS remains poor' - please provide some references. With current treatment methods, inlcuding biologicals, is this really the case?
Response:Thank you for your good advice. And we have added some references in the revised manuscript.
(kiyama K, Samukawa Y, Hoshikawa H. Early postoperative endoscopic score can predict the long-term endoscopic outcomes in eosinophilic chronic rhinosinusitis (ECRS) patients. Braz J Otorhinolaryngol. 2023 Jan-Feb;89(1):136-143.
Yilmaz GO, Cetinkaya EA, Eyigor H, et al. The diagnostic importance of periostin as a biomarker in chronic rhinosinusitis with nasal polyp. Eur Arch Otorhinolaryngol. 2022 Dec;279(12):5707-5714.
Gomes SC, Cavaliere C, Masieri S, et al. Reboot surgery for chronic rhinosinusitis with nasal polyposis: recurrence and smell kinetics. Eur Arch Otorhinolaryngol. 2022 Dec;279(12):5691-5699. ).
(4)Line 81-82: Please provide the name of the institution that has issued the informed
consents.
Response: Thanks for your comments, which is highly appreciated. Just as you mentioned,all patient datas were from The First Affiliated Hospital of Nanchang University. And we have added in the revised manuscript in Line 88.
(5)Line 74 states that 20 subjects with CRS were included. Please elaborate how many of those were with and how many without polyps.
Response:Thank you for your suggestion.The number of chronic rhinosinusitis with nasal polyps (CRSwNP) is 10 , and the number of chronic rhinosinusitis without nasal polyps (CRSwNP) is 10 in Line 86-87.
- Line 21 states that ''A normal nasal mucosa 21 tissue culture model was established in vitro''. However further in the discussion authors mention that patients with CRS were Please clarify.
Response:In the method, we use normal tissue for cell culture in vitro. The patients with CRS were included, and were used for western blot in our study, the result is Figure 1. We used IL-13 to stimulate HNECs in vitro and explore the mechanism of type 2 inflammation in CRS. We only examined the expression of relative factors in CRS and did not use the tissue of CRS to establish the cell culture model.

Reviewer 2 Report
The article addresses a very interesting topic, chronic rhinosinusitis being a very diffuse pathology. The authors test, in an ex-vivo system, the response of nasal epithelium cells to interleukin 13, and the modulation of certain proteins related to the inflammatory process.
However, the article has serious limitations, which I list below:
the introduction is confusing, shallow, and not always relevant to the topic of the paper
the materials and methods are inadequately described and in need of extensive and accurate redrafting
many inaccuracies are written in the results. The experimental aspect is well designed but I suggest the addition of a viability test to strengthen the results presented
I list below some more detailed comments that I hope will help the authors to improve their manuscript
Abstract
Line 31: what is the CRS group?
Introduction
the introduction is full of assertions, more or less consistent with the topic of the paper, but which are not elaborated or explained in depth
Lines 38-39: does the presence or absence of nasal polyps have anything to do with the topic of the work?
Line 40: CRS has a significant impact on the quality of life of patients and imposes a large social burden. Which social burden?
Line 41: The prognosis of CRS is affected by several factors, especially the pathophysiological mechanism. What does this mean? Which is the pathophysiological mechanism affecting the prognosis?
Line 45: what is eCRS?
What is mucin and what’s its role?
Line 57-58 “These studies thus demonstrate that MUCs play an important role in the pathogenesis of CRS.” as they are described, these studies only show that in patients with crs MUC5AC and MUC5B are more highly expressed than in controls
Line 61: what is pendrin?
Lines 68-69: through interleukin (IL)-13 stimulation, we knocked down pendrin in HNECs, really? I don't agree, this is not what you have done.
M&M
Line 81: The study was approved by the Medical Research Ethics Committee of xxx,
please describe in more detail the population included in the study
Line 87: please specify the composition of the Digestive enzyme mixture. How long does the digestion process take? How was the cell population derived from the isolation characterized?
Line 89-92: please be more precise in your description of the protocol, it seems to me that the process thus described is difficult to replicate
Line 96: how were cell lysates prepared?
Line 124: How much protein was loaded for each sample?
Line 134: please provide more information about the development stage (membranes were exposed in a dark room is not enough)
Results
Line 145: Author refer to “the CRSsNP and CRSwNP” but when they described the population included in the study they only referred to 20 CRS patients. Please explain. Moreover, we do not know how many subjects each group consists of, and we do not know, much more importantly, whether mucosal sampling in patients with nasal polyps was done in an area with polyps, near it, or in the normal mucosa
Fig 1A: what’s the difference between the lanes 1-3, or 4-6 and so on? Are samples deriving from different patients? Please specify
Line 155: at concentrations < 50 ng/mL, pendrin, MUC5AC, and MUC5B mRNA were significantly upregulated in primary cultured HNECs this is not what is shown in Figure 2, where by the way, the significances are not shown. I would ask the authors to demonstrate with a viability assay that culturing cells in the presence of IL13 at the concentrations tested does not change cell numbers. This finding would be useful in demonstrating that the increase in protein expression is not due simply to an increase in cell numbers.
Moreover, perhaps the authors could explain to non-mucin experts why mucin is quantified by ELISA in cell supernatant, while other proteins are quantified by WB
Fig.3: Please show the significance compared with the control,
Line 172: please provide more details on the siRNA used
Figure 4: FAM NC is not shown, despite what is written in the caption. Or, be more precise/clear
Figure 5: After transfection of HNECs with siRNA pendrin-920, the protein expression of pendrin was downregulated ONLY in 1650!. Moreover, why in the figure some significances are indicated with p and others with an asterisk? The same in figure 6.
how do the authors explain that pendrin mRNA reduction is observed in all siRNAs while protein expression reduction is observed only with siRNA 1650?
Lines 194: “mRNA and protein levels of HNE and EGFR expression decreased” in which conditions?
Lines 195-197: These results indicate that pendrin may affect the mucin expression of HNECs by activating HNE/EGFR, I disagree, these results indicate that pendrin inhibition decreases expression of HNE and EGFR, but give no mechanistic indication
The same sentence “Pendrin is mainly expressed in the inner ear, thyroid, and kidney” in line 61 and 219
Line 229: (Figure 7), What is it referring to?
Author Response
Dear reviewer,
Thank you for your comments concerning our manuscript entitled “Effects of pendrin protein in nasal epithelial cells on mucin production in the context of type 2 inflammation” (ID: jpm-2164344). Those comments are all valuable and very helpful for revising and improving our paper and, as well as the important guiding significance to our researches. We have studied comments carefully and have made correction which we hope meet with approval. Revised portion are marked in red in the manuscript. The main corrections in the paper and the responds to the reviewer’s comments are as flowing:
Abstract
(1)Line 31: what is the CRS group?
Response:The CRS group is the samples from 20 CRS patients, including 10 CRS patients with nasal polyps,10 CRS patients without nasal polyps.
Introduction
the introduction is full of assertions, more or less consistent with the topic of the paper, but which are not elaborated or explained in depth
(2)Lines 38-39: does the presence or absence of nasal polyps have anything to do with the topic of the work?
Response:According to EPOS 2020, we have a new CRS classification, and CRS is classified as primary chronic rhinosinusitis and secondary chronic rhinosinusitis.
(3) Line 40: CRS has a significant impact on the quality of life of patients and imposes a large social burden. Which social burden?
Response:Social burden is that recurrent surgical procedures of CRS need operation cost, missed workdays (absenteeism) and decreased productivity at work.
(4)Line 41:The prognosis of CRS is affected by several factors, especially the pathophysiological mechanism. What does this mean? Which is the pathophysiological mechanism affecting the prognosis?
Response:The recurrence rate of CRS is high, especially in CRSwNP. The high recurrence of CRS.The pathophysiological mechanism affecting the prognosis include inflammation, mucociliary,dysfunction and changes in the microbial community interact with differing influences.
(5)Line 45: what is eCRS?
Response:Eosinophilic CRS is eCRS. Eosinophilic CRS (eCRS) requires quantification of the numbers of eosinophils, i.e. number/high powered field (HPF (400x) and EPOS2020 supports 10 or >/HPF.
(6)What is mucin and what’s its role?
Response:Mucins (MUC) are the major components of mucus, and secretion of large amounts of MUC can reduce movement of the cilia on epithelial cells and damage the respiratory defense function, resulting in repeated secondary bacterial infections and uncontrolled respiratory diseases.
- Line 57-58 “These studies thus demonstrate that MUCs play an important role in the pathogenesis of CRS.” as they are described, these studies only show that in patients with crs MUC5AC and MUC5B are more highly expressed than in controls.
Response:Thank you for good suggestion.We have corrected the expression of the sentences, as “These studies thus demonstrate that MUCs may play an important role in the pathogenesis of CRS.”
- Line 61: what is pendrin?
Response: Pendrin protein is a soluble transporter.An important part of the family 26A (SLC26A) member, encoded by SLC26A4 gene, can mediate cell chloride ion , iodine dissociation and the exchange of various anions such as zinc and bicarbonate ion to maintain steady state of medium ions in the tissue.And we have added the sentences in the revised manuscript.
- Lines 68-69: through interleukin (IL)-13 stimulation, we knocked down pendrin in HNECs, really? I don't agree, this is not what you have done.
Response: We can’t agree with you more. We have corrected the sentences in the revised manuscript. According to Sangon Biotech siRNA, the pendrin in HNECs was knocked down.
M&M
- Line 81: The study was approved by the Medical Research Ethics Committee of xxx,please describe in more detail the population included in the study
Response: The study was approved by the Medical Research Ethics Committee of The First Affiliated Hospital of Nanchang University
- Line 87: please specify the composition of the Digestive enzyme mixture. How long does the digestion process take? How was the cell population derived from the isolation characterized?
Response:The composition of the Digestive enzyme mixture include MEM 10ml, pancreatic enzymes 1.4mg/ml and Dnase 0.01mg/ml. The digestion process takes an hour. Epithelial cells of nasal mucosa are adherent cells and grow like pebbles.
- Line 89-92: please be more precise in your description of the protocol, it seems to me that the process thus described is difficult to replicate
Response:Thanks for your good advice.We have given more precise details in our description of the protocol for Primary Culture of Nasal Epithelial Cells in the revised manuscript.
- Line 96: how were cell lysates prepared?
Response:The cell lysates for protein are as follows. Remove the culture medium and drugs, gently rinse the adherent cells with precooled PBS for 3 times, pay attention to gentleness to prevent the cells from falling off, suck the remaining PBS solution in the culture dish for the last time, and discard it, add protein lysate and phosphorylated protease inhibitor, blow repeatedly, put them on the ice for 10 minutes, and then use the cell scraper to scrape the cells and lysate, and transfer them to 1.5ml EP tube, blow repeatedly to avoid gas. Bubble is generated, but attention should be paid to avoid the generation of foam, and it should be placed on ice to continue cracking for 20 minutes;Put the EP tube into a precooled 4°C centrifuge, 12000 g, centrifuge for 10 min, and take the supernatant.
The cell lysates for RNA are as follows. The cells were washed by D-hankS solution three times. After trypsin digestion, the cell suspension was collected with a 15ml centrifuge tube. Total RNA was extracted from cells with Trizol™ reagent (Cat No: 15596026, Invitrogen, Thermo Fisher Scientifific, USA).
- Line 124: How much protein was loaded for each sample?
Response:20µl protein was loaded for each sample.
- Line 134: please provide more information about the development stage (membranes were exposed in a dark room is not enough)
Response:Under the condition of avoiding light, the A and B solution (Beyotime Biotech,China) in the luminescent liquid reagent shall be mixed evenly by 1:1. After incubation for an appropriate time, membranes were exposed in the chemiluminescence instrument(Bio-rad company, USA) . We have revised the context in the revised manuscript.
Results
- Line 145: Author refer to “the CRSsNP and CRSwNP” but when they described the population included in the study they only referred to 20 CRS patients. Please Moreover, we do not know how many subjects each group consists of, and we do not know, much more importantly, whether mucosal sampling in patients with nasal polyps was done in an area with polyps, near it, or in the normal mucosa.
Response:In line 91-92, we have described “A total of 40 patients (10 CRS patients with nasal polyps,10 CRS patients without nasal polyps, and 20 nasal septum deviation patients) were included in this study from The First Affiliated Hospital of Nanchang University.” The normal nasal mucosa is from inferior turbinate, the CRS with nasal polyps was done in an area with polyps, and the CRS without nasal polyps was done in processus uncinatus.
- Fig 1A: what’s the difference between the lanes 1-3, or 4-6 and so on? Are samples deriving from different patients? Please specify
Response:The lanes 1-3 are the samples from the NC group, 4-6 are the samples from the CRSsNP patients and 7-9 are the aples from the CRSwNP patients. The samples are from different patients.
- Line 155: at concentrations < 50 ng/mL, pendrin, MUC5AC, and MUC5B mRNA were significantly upregulated in primary cultured HNECs this is not what is shown in Figure 2, where by the way, the significances are not shown. I would ask the authors to demonstrate with a viability assay that culturing cells in the presence of IL13 at the concentrations tested does not change cell numbers. This finding would be useful in demonstrating that the increase in protein expression is not due simply to an increase in cell numbers.
Response:It is a good suggestion. In our further study,we will add a viability assay.
- Moreover, perhaps the authors could explain to non-mucin experts why mucin is quantified by ELISA in cell supernatant, while other proteins are quantified by WB.
Response:We have tried to quantify mucin by WB,but we did not detect the expression of mucin. The reason may be the molecular weight of mucin is relatively large (predicted molecular weight: 527 kDa).
- 3: Please show the significance compared with the control,
Response:Thank you for your advice, but our purpose is to show the appropriate concentration of IL-13 is 50 ng/ml.
- Line 172: please provide more details on the siRNA used
Response: The pendrin siRNA was designed by Sangon Biotech, including NC, pendrin-373, pendrin-920, pendrin-1650, and FAM NC.
- Figure 4: FAM NC is not shown, despite what is written in the caption. Or, be more precise/clear
Response: FAM NC was just used to evaluate transfection efficiency,which was not for pendrin expression.
- Figure 5: After transfection of HNECs with siRNA pendrin-920, the protein expression of pendrin was downregulated ONLY in 1650!. Moreover, why in the figure some significances are indicated with p and others with an asterisk? The same in figure 6.
Response:Accoding the WB result, the protein expression of pendrin was downregulated only in siRNA pendrin1650. The p value was listed, when p was >0 0.05, while when p < 0 .05 was marked with an asterisk.
- how do the authors explain that pendrin mRNA reduction is observed in all siRNAs while protein expression reduction is observed only with siRNA 1650?
Response:Because the translation of protein is complicated, so the expression of mRNA may be inconsistent with protein.
- Lines 194: “mRNA and protein levels of HNE and EGFR expression decreased” in which conditions?
Response: After IL-13 (50 ng/ml) stimulation and pendrin siRNA transfection, mRNA and protein levels of HNE and EGFR expression decreased by qRT-PCR and WB analyses.
- Lines 195-197: These results indicate that pendrin may affect the mucin expression of HNECs by activating HNE/EGFR, I disagree, these results indicate that pendrin inhibition decreases expression of HNE and EGFR, but give no mechanistic indication
Response: Thank you for your good suggestion. We just show that These results indicate that pendrin may affect the mucin expression of HNECs by activating HNE/EGFR. The sthdy just show that the relation between pendrin and HNE/EGFR, but in our further study, we will do mechanism between pendrin and HNE/EGFR.
- The same sentence “Pendrin is mainly expressed in the inner ear, thyroid, and kidney” in line 61 and 219.
Response:Thanks. We have deleted the sentence in the revised manuscript.
- Line 229: (Figure 7), What is it referring to?
Response: It is referring a positive relationship between pendrin and EFGR/HNE.

Round 2
Reviewer 1 Report
I thank the authors for ppropriately adressing my comments.
I have two minor remarks:
Line 19: and paranasal sinus -> 'paranasal sinuses'
Line 27: NC group -> please decipher the abbreviation
Author Response
Dear professor,
Thank you for your comments concerning our manuscript . We have studied comments carefully and have made correction which we hope meet with approval. Revised portion are marked in red in the manuscript. The main corrections in the paper and the responds to the reviewer’s comments are as flowing:
Line 19: and paranasal sinus -> 'paranasal sinuses'
Response:Thank you.We have corrected it in the revised manuscript.
Line 27: NC group -> please decipher the abbreviation
Response:NC is negative control.
Reviewer 2 Report
while some modifications to the introduction and to the materials and methods section were made, the authors did not take into account the modifications that were requested in the results section. the authors answered my questions in the answer to reviewers letter but I imagine that a letter might have the same curiosities, so I ask them to integrate the answers into the text
the captions for figures 1 to 7 are missing
the significances in fig. 3 have not been shown against the controls
Author Response
Dear professor,
Thank you for your good advice.
1,the captions for figures 1 to 7 are missing
Response: We have added the captions for figures.
2,the significances in fig. 3 have not been shown against the controls.
Response: We agree with you. The significances in fig. 3 have been shown against the controls.